# Effects of High Hydrostatic Pressure on the Distribution of Oligosaccharides, Pinitol, Soysapapogenol A, and Fatty Acids in Soybean

**DOI:** 10.3390/foods13142214

**Published:** 2024-07-14

**Authors:** Shigeaki Ueno, Hsiuming Liu, Risa Kishino, Yuka Oshikiri, Yuki Kawaguchi, Akio Watanabe, Wataru Kobayashi, Reiko Shimada

**Affiliations:** 1Faculty of Education, Saitama University, Saitama 3300061, Japan; 2Department of Food Science, National Taiwan Ocean University, Keelung 202301, Taiwan; 3Department of Food Science, Jumonji University, Saitama 3528510, Japan; akio-wa@jumonji-u.ac.jp; 4Department of Health and Nutrition Sciences, Komazawa Woman’s University, Tokyo 2068511, Japan; w-kobayashi@komajo.ac.jp

**Keywords:** soybean, high hydrostatic pressure, pinitol, oligosaccharide, soyasapogenol A, fatty acid

## Abstract

The effects of high hydrostatic pressure (HHP) treatment (100–600 MPa for 10–60 min) and thermal treatment (boiling for 10–60 min) on oligosaccharides, pinitol, and soyasapogenol A as taste ingredients in soybean (*Glycine max* (L.) Merr.) (cv. Yukihomare) were evaluated. Additionally, soybean-derived fatty acids such as α-linolenic acid, linoleic acid, oleic acid, palmitic acid, and stearic acid in pressurized soybeans were quantitatively analyzed. Sucrose, stachyose, and raffinose concentrations were decreased in all tested pressure and time combinations; however, pinitol concentrations were increased by specific pressure and time combinations at 100–400 MPa for 10–60 min. While the soyasapogenol A content in boiled soybeans decreased with increasing boiling time, that of pressurized soybeans was altered by specific pressure and time combinations. At the lower pressure and shorter time combinations, the essential fatty acids such as α-linolenic acid and linoleic acid showed higher contents. Stearic acid and oleic acid contents of pressurized soybeans increased at mild pressure levels (300–500 MPa). In contrast, the combination of higher pressure and longer time results in lower essential fatty acid contents. Non-thermal-pressurized soybeans have the potential to be a high-value food source with better taste due to the enrichment of low molecular weight components such as pinitol, free amino acids, and the reduction of isoflavones and Group A soyasapogenol.

## 1. Introduction

High hydrostatic pressure (HHP) technology is a novel modification method for physicochemical qualities as well as a non-thermal method for inactivating microorganisms in food products [1,2,3,4,5,6]. HHP alters the intercellular structure, affecting how proteins and small molecules interact and therefore stimulating specific enzymatic reactions [6,7,8]. HHP can induce various phase transitions of lipids in foods that lead to the solid–liquid phase transition of oils and the gel–liquid crystal transition of the double-layered membrane of phospholipids [1]. In other words, the HHP method reduces damage to food components such as color, flavor, and nutrition, while also inactivating bacteria to produce high-quality and safe foods.

Soybean (*Glycine max* (L.) Merr.) is well known as a commercially available, nutritional source of low-cost proteins, amino acids, and lipids [9,10]. In Asian countries, soybeans are an essential and traditional food source, including tofu, soymilk, soy sauce, miso, natto, soy paste, and tempeh. Recently in Japan, soymilk has been widely used as an alternative to cow milk. On the other hand, in Western countries, soybeans are mainly processed into soybean meal and seed oil [10]. Soybean seed oil contains abundant fatty acids such as linoleic acid (18:2), followed by oleic acid (18:1), palmitic acid (16:0), linolenic acid (18:3), and stearic acid (18:0). Unsaturated fatty acids such as oleic, linoleic, and linolenic acids have several positive effects on human health, such as prevention of atherosclerosis, reduction of total and low-density lipoprotein cholesterol and triacylglycerol levels in plasma, and suppression of inflammatory processes [10].

Dried soybean is generally edible after water-soaking and boiling, which leads to a softened texture and reduction in taste-related compounds such as lipoxygenase. On the other hand, certain enzymatic reactions occur during the processing and storage of soybeans, resulting in changes in ingredients such as proteins, polysaccharides, and lipids. HHP has made it possible to soften the texture and enrich free amino acids via a quick enzymatic reaction at optimum pressure and time combinations; this is especially beneficial for the nutritional components of soybeans [11,12,13]. Applying HHP at 200 MPa for 10 min increased the amount of free amino acids during subsequent storage via a slow enzymatic reaction [13]. HHP is also regarded as a promising technology with minimal impact on food products’ sensory attributes, promoting inhibition of the enzymatic oxidation catalyzed by specific enzymes such as polyphenol oxidase [14,15,16].

In a previous study, we investigated how enriched free amino acids and decreased astringent taste components such as isoflavones in pressurized soybeans significantly improved sensory perception [17]. While unpressurized soybeans had a bitter and astringent taste, these tastes disappeared and the umami-sweet taste appeared in soybeans pressurized at over 500 and 600 MPa. The decrease in astringent taste compounds was partly caused by the decrease in isoflavones, such as genistein-, daizein-, and glycitein-type molecules, by HHP. Although isoflavone is one of the astringent–bitter taste compositions, other astringent–bitter taste compositions such as saponin were not measured in our previous study [17].

Additionally, it is possible that the increase in sweet components such as oligosaccharides reduces the intensity of astringent–bitter taste. However, it is still unclear how other crucial taste components, such as oligosaccharides (sweet), pinitol, and soyasaponin A (bitter/astringent), are distributed throughout pressurized soybeans. Furthermore, knowledge of the distribution of fatty acids in soybeans is important from the viewpoint of the commercial value of soybean oils; however, the effects of HHP on the fatty acids in soybeans are still unknown.

The purpose of this study was to compare the distributions of key oligosaccharides, such as sucrose, stachyose, and raffinose, as well as a cyclic polyol called pinitol, in soybeans subjected to 100–600 MPa pressure for 10, 30, and 60 min, with soybeans under thermal treatment. Furthermore, because soyasaponin A is one of the low molecular weight components that contribute to the bitter–astringent taste, the distribution of soyasaponin A in pressurized and thermally treated soybeans was investigated to better understand pressure-assisted taste alterations. Soybean-derived fatty acids are commercially valuable; therefore, selected fatty acids, including α-linolenic acid, linoleic acid, oleic acid, palmitic acid, and stearic acid, in pressurized soybeans were also analyzed.

## 2. Materials and Methods

### 2.1. Materials

Soybeans (cv. Yukihomare) harvested in Hokkaido, Japan, in November 2021 and October 2022 (Asahishokuhin Corp., Kobe, Japan) were purchased from a local supermarket. Soyasapogenol A was purchased from Koshiro Company Ltd. (Osaka, Japan) and the fatty acid methylation kit was purchased from Nacalai Tesque Inc. (Kyoto, Japan). Methyl ester standards and methyl heptadecanoate were purchased from Sigma Aldrich Inc. (St. Louis, MO, USA). All other reagents were purchased from Fujifilm Wako Pure Chemical Corporation (Osaka, Japan).

All reagents from Fujifilm Wako Pure Chemical were special grade, except for acetonitrile, which was HPLC grade. Ultrapure water was generated using a Simplicity UV system (Millipore, Belford, MA, USA).

### 2.2. Preparation of Soybeans

Approximately 20 g of dried soybeans (cv. Yukihomare) were soaked in 60 g of ultrapure water for 15 h. The moisture content of dried soybean before water-soaking was approximately 0.22 g-water/g-d.s., and that of water-soaked soybean was approximately 1.90 g-water/g-d.s. In this present study, all analyses were carried out with the water-soaked soybeans. After water soaking, approximately 50 g of soybeans were wiped with a paper towel and vacuum-packed in polyethylene pouches (Eiken Kizai, Tokyo, Japan) using a vacuum sealer (Furukawa Seisakujo, Tokyo, Japan). The soybean samples in pouches were placed in a stainless-steel vessel of an HHP device (MFP7000; Mitsubishi Heavy Industry, Tokyo, Japan). At 25 °C, the pressure in a sample vessel was adjusted to 100, 200, 300, 400, 500, and 600 MPa for 10, 30, and 60 min. Ultrapure water was used as a hydraulic fluid in the stainless-steel vessel [15,17]. The temperature in the vessel was maintained by circulating water in the outer shell of the vessel, and the temperature of the hydrostatic fluid in the vessel reached up to 35 °C, as monitored by a thermocouple.

Water-soaked and unpacked soybeans not subjected to HHP were immersed in boiling water for 60 min at 10 min intervals. After boiling, approximately 50 g of soybeans were wiped with a paper towel and vacuum-packed in pouches. After pressure and boiling treatments, pouches of unpressurized, pressurized, and boiled soybeans (approximately 50 g of soybeans) were frozen immediately at −80 °C until further examination. At least 15 biologically different soybean grains were used for the single analysis, and all quantification analyses were carried out in triplicate. Boiled soybeans were used for pinitol, oligosaccharides, and soyasapogenol A analyses but not for fatty acid analyses.

### 2.3. Determination of Pinitol, Oligosaccharides, and Soyasapogenol A

#### 2.3.1. Pinitol and Oligosaccharides Extract Sample Preparation

One gram of frozen soybeans (cv. Yukihomare) was cut into small particles with a knife, and then soybean particles with 5 g of ultrapure water were homogenized with Polytron MR2100 (Kinematica, Malters, Switzerland) at 12,000 rpm for 5 min at 4 °C. The obtained soymilk was centrifuged at 12,000× *g* for 15 min at 4 °C (himac CR15, Hitachi Corp., Tokyo, Japan). After centrifugation, 500 μL of supernatant was filtrated with an ultrafiltration unit (Amicon Ultra-0.5 mL-10 kDa, Merck, MO, USA). The resulting filtrates were applied to further HPLC analyses.

#### 2.3.2. Quantification of Pinitol and Oligosaccharides

Pinitol and oligosaccharides such as raffinose, stachyose, and sucrose in soybeans (g/100 g of soybean w.b.) were measured using an HPLC equipped with a RID detector (RID-10A, Shimadzu Corp., Kyoto, Japan) and column (Shodex SH1821, Showa Denko, Tokyo, Japan) [18]. The separation column was combined with the guard column (Shodex SH-G, Showa Denko, Tokyo, Japan) and column temperatures were maintained at 25 °C in the column oven (CTO-10AC VP, Shimadzu Corp., Kyoto, Japan). A total of 50 μL of soybean extracts was injected automatically with an Auto-Injector (SIL-6B, Shimadzu Corp., Kyoto, Japan). The pinitol and oligosaccharides eluent was 5 mM H_2_SO_4_ with a flow rate of 1 mL per min. Quantification data from the RID detector were stored in the data acquisition software (Chromato pro version 4.0, Runtime Instrument Co., Ltd., Tokyo, Japan).

#### 2.3.3. Soyasapogenol A Extract Sample Preparation

The soyasapogenol A extraction method was described by Tsukamoto et al. [19]. Soybean samples (cv. Yukihomare) were frozen at −20 °C for 24 h and continuously vacuumed for 24 h. Semi-dried soybean samples were then ground with a coffee mill (Ultra centrifugal mill, Miura Riken Kougyou, Tokyo, Japan). A total of 600 mg of the ground soybean powder was vortexed intensely with 6.6 mL of 70% ethanol containing 0.1% acetic acid and then placed in an incubator (BR-23FP, Titec, Tokyo, Japan) at 25 °C for 48 h. After incubation, the soyasapogenol A soybean sample extract was centrifuged at 20,000× *g* for 10 min at 4 °C (himac CR15, Hitachi Corp., Tokyo, Japan). After centrifugation, 1.5 mL of supernatant was filtered (0.22 μm, Star Lab Scientific, Tokyo, Japan) and vortexed with 150 μL of hydrochloric acid in a microtube. The microtube was fixed in an aluminum rack and transferred into a water bath (SB-350, EYELA Water bath, Tokyo Rika Kiki Corp., Tokyo, Japan). Samples were heated at 80 °C for 6 h to facilitate hydrolysis. After hydrolysis, samples were cooled to room temperature and hydrolyzed soybean extract samples were used for further soyasapogenol A analyses using HPLC.

#### 2.3.4. Quantification of Soyasapogenol A

Soyasapogenol A analyses were carried out using the other HPLC system. After hydrolysis, the contents of soyasapogenol A (μg/g of soybean w.b.) were measured using HPLC with a UV-VIS detector (SPD-20A, Shimadzu Corp, Kyoto, Japan) and ODS column (Capcellpak C-18 AG-120, Shiseido, Tokyo, Japan) at 210 nm [19]. The gradient mobile phase was used for the quantification of soyasapogenol A. The mobile phase A was ultrapure water and acetic acid (100:0.1, *v*/*v*), and the mobile phase B was acetonitrile and acetic acid (100:0.1, *v*/*v*). The gradient condition of mobile phase B was initially 65%. After injection of 50 μL of the sample solution using an auto-injector (SIL-10AD vp, Shimadzu Corp., Kyoto, Japan), the gradient system was used to quantify soyasapogenol A. The mobile phase B was changed to 75% for the first 12.5 min and maintained for the next 1.5 min (12.5–13 min), changed to 100% (at 13.1–15 min), and finally back to the initial condition of 65% (15.1–30 min). The concentration of soyasapogenol A (aglycon) was taken as the total concentration of Group A soyasaponins (glycoside). Quantification data from the UV-VIS detector were stored in the data acquisition software (Chromato pro version 4.0, Runtime Instrument Co., Ltd., Tokyo, Japan).

### 2.4. Determination of α-Linolenic Acid, Linoleic Acid, Oleic Acid, Palmitic Acid, and Stearic Acid

#### 2.4.1. Fatty Acids Extract Sample Preparation

A total of 10 g of untreated and pressurized soybeans (cv. Yukihomare) was cut into small particles with a knife and mixed with 60 mL of chloroform:methanol (2:1, *v*/*v*). The mixture was homogenized at 10,000 rpm for 5 min at 4 °C with Polytron MR2100, and incubated at 65 °C for 60 min. The obtained solution was concentrated using a rotary evaporator (N-1300E, EYELA Water bath, Tokyo Rika Kiki Corp., Tokyo, Japan). The obtained concentrates were applied to further pretreatments, such as methylation, according to the manufacturer’s instruction manual. Concentrated samples were subjected to methylation using the fatty acid methylation kit (Nacalai Tesque Inc., Kyoto, Japan). Briefly, the concentrated sample was mixed with 500 μL of methylation solution A (toluene 52% and methanol 48%) and reacted with 500 μL of methylation solution B, containing 93% methanol, at 37 °C for 60 min, and subsequently, with 500 μL of methylation solution C containing 30% methanol. Finally, 1 mL of isolation reagent was added and the upper phase solution was mixed with 1 mL of ultrapure water. The obtained solution was applied to a GC-MS to quantify fatty acids.

#### 2.4.2. Quantification of α-Linolenic Acid, Linoleic Acid, Oleic Acid, Palmitic Acid, and Stearic Acid

The distributions of fatty acids such as α-linolenic acid, linoleic acid, oleic acid, palmitic acid, and stearic acid were measured (g/100 g of soybean w.b.) using a GC-MS (GCMS-QP 2020NX, Shimadzu Corp., Kyoto, Japan) [20]. Samples for quantification were diluted using hexane for further GC-MS analyses. A volume of 1 µL of methylation samples in vials was injected into the column (DB-23, Agilent Technologies, Santa Clara, CA, USA). The analyses were carried out in the SIM mode, and the injection method was split mode at a split ratio of 10:1. The initial oven temperature was maintained at 60 °C for 1.5 min, increased at 40 °C per min to 200 °C and maintained here for 3 min, and then increased at 25 °C per min to 250 °C and held here for 1 min. The inlet and detector temperatures were maintained at 250 °C. The ion source temperature was maintained at 200 °C during analysis. The flow rate of the column’s helium carrier gas was 5.0 mL per minute. Peaks in the chromatograms were identified using 37 methyl ester standards and quantified using the internal standard methyl heptadecanoate.

### 2.5. Statistical Analysis

To verify the results of this present study, all the experiments were carried out in triplicate and means ± standard deviations were calculated. Analyses were conducted using the software JMP Pro 12.2 (SAS Inc., Cary, NC, USA). The Tukey–Kramer method was used to determine statistical significance and differences at *p* < 0.05 were considered significant.

## 3. Results

### 3.1. Pinitol and Oligosaccharide Concentrations in Pressurized Soybean

The distributions of pinitol and three oligosaccharides—sucrose, stachyose, and raffinose—were measured in untreated and pressurized soybeans (cv. Yukihomare) (Table 1). Figure 1 shows the contour diagrams of pinitol, sucrose, stachyose, and raffinose concentrations for all pressure and time combinations. Pinitol, sucrose, stachyose, and raffinose concentrations in unpressurized soybeans were 2.52 ± 0.54, 14.69 ± 2.43, 5.88 ± 1.27, and 1.82 ± 0.49 g/100 g, respectively.

The highest concentrations of pinitol, sucrose, stachyose, and raffinose in pressurized soybeans were 5.48 ± 0.07 g/100 g (at 200 MPa and 10 min), 13.35 ± 0.74 g/100 g (at 200 MPa and 10 min), 5.78 ± 0.56 g/100 g (at 400 MPa and 30 min), and 1.70 ± 0.02 g/100 g (at 100 MPa and 60 min), respectively. In contrast, the lowest concentrations of pinitol, sucrose, stachyose, and raffinose were 1.16 ± 0.10 g/100 g (at 500 MPa and 60 min), 4.81 ± 0.10 g/100 g (at 500 MPa and 30 min), 2.75 ± 0.13 g/100 g (at 600 MPa and 30 min), and 0.55 ± 0.05 g/100 g (at 600 MPa and 10 min), respectively (Table 1). Sucrose, stachyose, and raffinose concentrations decreased in all tested pressure and time combinations; however, pinitol concentrations increased at the specific pressure and time combinations evaluated at 100–400 MPa for 10–60 min (except at 400 MPa for 60 min). Significant differences were observed at the various pressure and time combinations.

Overall, the pressure and time combinations tested in this study (100 MPa for 10 min and 600 MPa for 60 min) resulted in lower sucrose, stachyose, and raffinose concentrations (Figure 1). In contrast, treatments ranging from 100 MPa for 10 min to 400 MPa for 30 min resulted in enriched pinitol concentrations, for example, a 2.2-fold increase at 200 MPa for 10 min compared to untreated (unpressurized) soybeans (Figure 1).

### 3.2. Soyasapogenol A Concentration in Pressurized Soybean

The distribution of soyasapogenol A as an astringent–bitter taste component was measured in untreated and pressurized soybeans (cv. Yukihomare) (Table 1). Figure 2 shows the contour diagram of soyasapogenol A concentrations under all pressure and time conditions. Soyasapogenol A concentration in unpressurized soybeans was 429.53 ± 13.07 μg/g. The highest concentration in pressurized soybeans was 502.89 ± 7.69 μg/g (at 500 MPa and 10 min), while the lowest concentration was 262.73 ± 9.88 μg/g (at 200 MPa and 60 min).

Overall, most pressure and time combinations between 100 MPa for 10 min and 600 MPa for 60 min reduced soyasapogenol A (Figure 2) concentrations. In contrast, specific pressure and time combinations, including 300 MPa for 60 min, 400 MPa for 60 min, 500 MPa for 10 and 30 min, and 600 MPa for 60 min, resulted in significant enrichment of soyasapogenol A compared to untreated soybeans (*p* < 0.05) (Figure 2). Significant differences were observed at the various pressure and time combinations.

### 3.3. Pinitol, Oligosaccharides, and Soyasapogenol A Concentrations in Boiled Soybean

The distributions of pinitol, sucrose, stachyose, raffinose, and soyasapogenol A in boiled soybeans (cv. Yukihomare) were also measured (Table 2). The concentrations of pinitol, sucrose, stachyose, raffinose, and soyasapogenol A in boiled soybeans ranged from 2.92 ± 0.13 to 3.66 ± 0.30 g/100 g, from 8.91 ± 0.01 to 10.90 ± 0.36 g/100 g, from 3.37 ± 0.09 to 4.22 ± 0.15 g/100 g, from 1.34 ± 0.07 to 1.50 ± 0.06 g/100 g, and from 279.15 ± 1.17 to 363.17 ± 1.09 μg/g, respectively. Overall, boiling significantly decreased the quantities of sucrose and stachyose (*p* < 0.05), and slightly decreased raffinose concentrations, while significantly increasing pinitol concentrations (*p* < 0.05). Pinitol concentrations in boiled soybeans resulted in an enrichment of 1.4-fold compared to untreated (unpressurized) soybeans (Table 2). Soyasapogenol A concentrations in boiled soybeans significantly decreased with increasing boiling time (*p* < 0.05).

### 3.4. Fatty Acid Concentrations in Pressurized Soybean

The distributions of major fatty acids such as α-linolenic acid, linoleic acid, oleic acid, palmitic acid, and stearic acid were measured in untreated and pressurized soybeans (cv. Yukihomare) (Table 3). Figure 3 shows the contour diagram of fatty acid concentrations at all pressure and time combinations. Fatty acid concentrations in untreated soybeans were 1.73 ± 0.35 g/100 g-w.b. for α-linolenic acid, 5.62 ± 0.43 g/100 g-w.b. for linoleic acid, 1.44 ± 0.15 g/100 g-w.b. for oleic acid, 0.31 ± 0.03 g/100 g-w.b. for palmitic acid, and 0.06 ± 0.01 g/100 g-w.b. for stearic acid.

The highest concentrations of α-linolenic acid, linoleic acid, oleic acid, palmitic acid, and stearic acid in pressurized soybeans were 2.57 ± 1.11 g/100 g-w.b. (at 100 MPa and 10 min), 8.02 ± 2.08 g/100 g-w.b. (at 100 MPa and 10 min), 2.53 ± 0.84 g/100 g-w.b. (at 400 MPa and 30 min), 0.34 ± 0.04 g/100 g-w.b. (at 100 MPa and 10 min), and 0.10 ± 0.06 g/100 g-w.b. (at 400 MPa and 30 min), respectively. In contrast, the lowest concentrations of α-linolenic acid, linoleic acid, oleic acid, palmitic acid, and stearic acid in pressurized soybeans were 1.25 ± 0.46 g/100 g-w.b. (at 600 MPa and 10 min), 4.75 ± 1.07 g/100 g-w.b. (at 600 MPa and 10 min), 1.35 ± 0.10 g/100 g-w.b. (at 300 MPa and 10 min), 0.22 ± 0.03 g/100 g-w.b. (at 500 MPa and 30 min), and 0.06 ± 0.01 g/100 g-w.b. (at 100 MPa and 60 min, 200 MPa and 60 min, and 300 MPa and 10 min), respectively (Table 3).

Overall, the pressure and time combinations tested in this study (from 100 MPa for 10 min to 600 MPa for 60 min) resulted in different contour diagram patterns for fatty acids (Figure 3). The relative contents of fatty acids such as α-linolenic acid, linoleic acid, oleic acid, palmitic acid, and stearic acid in pressurized soybeans ranged from 0.72 to 1.5, from 0.85 to 1.4, from 0.93 to 1.8, from 0.70 to 1.1, and from 1.1 to 1.6, respectively. Compared with the untreated soybeans, specific pressure and time combinations led to a higher accumulation of fatty acids. The higher pressure resulted in the reduction of fatty acids such as α-linolenic acid, linoleic acid, linoleic acid, and palmitic acid, but not stearic acid, in pressurized soybeans (Figure 3). In contrast, the stearic acid content in pressurized soybeans increased at higher pressure (400–500 MPa (Figure 3). Significant differences were rarely found in the pressurized soybeans at all tested pressure and time combinations.

Significant differences in α-linolenic acid content in pressurized soybeans were observed at three pressure and time combinations: 200 MPa × 10 min vs. 300 MPa × 60 min, 200 MPa × 30 min vs. 400 MPa × 30 min, and 500 MPa × 60 min. For linoleic acid, significant differences were observed at 100 MPa × 60 min vs. 500 MPa × 60 min, 200 MPa x30 min vs. 400 MPa × 30 min and 500 MPa × 60min, and 400 MPa × 60 min vs. 600 MPa × 60 min. As for oleic acid content, only one pressure and time combination (200 MPa × 30 min vs. 400 MPa × 30 min) showed a significant difference. Significant differences in palmitic acid content were observed at five pressure and time combinations: 100 MPa × 60 min vs. 500 MPa × 30 min, 200 MPa × 30 min vs. 400 MPa × 30 min and 500 MPa × 60 min, 200 MPa × 60 min vs. 400 MPa × 30 min, and 400 MPa × 60 min vs. 600 MPa × 60 min. For stearic acid, significant differences were observed at 100 MPa × 60 min vs. 500 MPa × 30 min, 200 MPa × 30 min vs. 400 MPa × 30 min and 500 MPa × 60 min, and 400 MPa × 60 min vs. 600 MPa × 60 min.

## 4. Discussion

The application of HHP to soybeans (cv. Yukihomare) at 100–600 MPa for 10–60 min resulted in a higher accumulation of pinitol and a reduction in oligosaccharides at specific pressure and time combinations. During seed maturation, oligosaccharides such as sucrose, stachyose, and raffinose accumulate in soybean embryos [21]. Of the sugars in soybeans, sucrose is the biggest component (48.90%, 50.65 mg/g), followed by stachyose (39.26%, 40.68 mg/g) and raffinose (9.38%, 9.762 mg/g), with glucose and fructose contributing very little (2.46%) [22]. In our study, among the measured components, sucrose was similarly the most abundant oligosaccharide in pressurized soybeans (48.98–65.25%), followed by stachyose (24.20–42.86%) and raffinose (6.11–9.55%), while unpressurized soybeans contained 65.61%, 26.27%, and 8.13% of sucrose, stachyose, and raffinose, respectively (Table 1). Boiling for 29–37 min and blanching at 96–98 °C for 3 min 15 s decreases sucrose, stachyose, and raffinose content in soybean [23], consistent with our findings on sucrose, stachyose, and raffinose concentrations, which decreased following boiling treatment (Table 2). As a result, these oligosaccharides were not significantly responsible for the pressure-assisted taste alternation, particularly the increase in sweet-tasting compounds. Although soybean contains soluble carbohydrates, primarily sucrose, raffinose, and stachyose, that act as sweet-tasting compounds, raffinose and stachyose are largely undigested in nonruminant animals [21]. Therefore, foods with low raffinose and stachyose levels are preferable. Based on this viewpoint, applying a higher level of HHP enabled the generation of low raffinose and stachyose soybean with relative concentrations (pressurized sample/unpressurized sample) significantly decreased to 0.30 for raffinose at 600 MPa for 10 min and 0.47 for stachyose at 600 MPa for 30 min (*p* < 0.05).

Environmental changes such as water stress from drought and high salinity cause plants to undergo osmotic adaptions, including increasing the number of osmoprotectants such as betaines, amino acids (Pro, ectoine), polyol, and sugars like D-ononitol, D-pinitol, and trehalose, and sustaining turgor pressure [24]. Because D-pinitol has a role in drought and high salinity stress as well as in heat-induced water deficits in legumes, pinitol concentrations are increased under stress conditions [25]. As previously stated, pinitol concentrations in pressurized soybeans increased by up to 2.2-fold that of unpressurized soybeans (Table 1). Furthermore, in our previous study, Pro concentrations in pressurized soybeans increased significantly, up to 0.38 μmol/g (400 MPa, 30 min), compared with concentrations in untreated soybeans (0.09 μmol/g) [11]. This high accumulation of pinitol and Pro in pressurized soybeans was comparable to levels occurring during drought and high salinity stress in legumes. Furthermore, D-pinitol has antioxidant, anti-inflammatory, and anti-carcinogenic properties [26] and improves diabetic sarcopenia [27]. D-pinitol also increases the expression of proteins with anti-inflammation and antiapoptosis functions [27]. Therefore, HHP, resulting in a high accumulation of pinitol, would be an alternative technique for investigating value-added soybeans.

Raw soybeans are not suitable for human consumption due to the presence of anti-nutritional components such as saponins, trypsin inhibitors, and allergens that are harmful to the consumer. Inactivating antinutritional factors in soybean meal enhances the retention and utilization of its components [28]. Group A soyasaponin had the highest undesirable intensity of soybean glycosides, which showed a weaker desirable taste tendency because of decomposition to their aglycones following thermal processing [29]. In this study, soyasapogenol A content in pressurized soybean was decreased by specific pressure and time treatments, comparable to boiled soybean, which was considerably reduced based on boiling time (*p* < 0.05). Kamo reported that soyasaponin A contents in soybean products ranged from 54.5 to 360 μmol/g w.b. in miso, from 163 to 239.8 μmol/g w.b. in natto, and from 84.5 to 133.1 μmol/g w.b. in soymilk [30]. In contrast, the concentration of soyasapogenol A, which is hydrolyzed from group A soyasaponin, was estimated at 470 μmol/g d.b. in untreated soybeans and 326–532 μmol/g d.b. in pressurized soybeans. Although it is still unclear which components cause a reduction in the bitter–astringent taste in pressurized soybeans, soyasapogenol A concentrations in specific pressure- and time-treated soybeans were at levels that were almost similar to those in commercial edible soybean products without the bitter–astringent taste. One possible reason for the alteration in the taste of soybean from a bitter–astringent taste in unpressurized soybean to a sweet taste in higher pressurized soybean is the reduction in soyasapogenol A. Compounds with bitter–astringent taste were covered with sugars and attenuated below the threshold level by powerful sweet-tasting compounds in several meals and medicines, allowing for a decrease in undesirable taste [29]. In our previous study, HHP facilitated the partial destruction of polysaccharides such as starch and fiber such as β-glucan [31]. Based on these findings, oligosaccharide concentrations in pressurized soybeans were reduced; thus, the reduction of the bitter–astringent taste in pressurized soybeans is unrelated to oligosaccharide concentrations.

Soybeans are a commercially available source of plant oils; thus, information on the contents and constituents of fatty acids in pressurized soybeans is important. Essential fatty acids such as α-linolenic acid and linoleic acid have high nutritional value and functional properties such as hypocholesterolemia, hypotriglyceridemic, and improved cardiovascular function [10]. Soy oil is packed into discrete subcellular structures called oil bodies or spheorosomes, which are in the cytoplasm of palisade-like cotyledon cells [10]. After the application of HHP at higher levels of over 400 MPa, cellular membrane structures would be partly destroyed, resulting in the oxidation of polyunsaturated fatty acids such as α-linolenic acid and linoleic acid. The higher pressure and the longer pressure time led to reductions in polyunsaturated fatty acids due to oxidation (Figure 3, Table 3). Zhang et al. reported that applying HHP to soy sauce accelerated the lipid oxidation and amino acid degradation induced by the Strecker reaction by increasing aldehydes and ketones [32].

In contrast, the highest contents of α-linolenic acid and linoleic acid were obtained at the lowest pressure and time combination of 100 MPa for 10 min. Considering the synthetic pathways and metabolic genes associated with major fatty acids in soybean, α-linolenic acid and linoleic acid are produced by the unsaturation of oleic acid following stearic acid production by the elongation of palmitic acid. The synthesis of these major fatty acids in soybeans has been reported to be catalyzed by ω-6 fatty acid unsaturation enzyme 2 (FAD2) and ω-3 fatty acid unsaturation enzyme 3 (FAD3) [10]. Slight enrichment of fatty acids can be caused by the optimization of enzymatic reactions after HHP-induced partial cellular destruction. In general, the oxidation of soybean-derived fatty acids produces a souring odor and the degradation of soybean-derived fatty acids causes a foul odor. Although quantitative odor evaluation was not performed, no sour or foul odor occurred under conditions of reduced linoleic and linolenic acids.

The mechanisms of HHP-induced nutritional changes were initially associated with oxidation of the cellular biomaterials, resulting in secondary metabolites at lower pressure [12]. However, at higher pressure (>200–300 MPa), HHP ruptures the cell membrane, increases the permeability of the cell wall, progressively terminates the metabolic activity of plant cells, and drastically diminishes cell viability. Moderate pressure accelerates the enzymatic reaction’s rate and catalytic function. HHP has been reported to disrupt cell membranes, allowing intracellular fluid to leak out of the cell and into the extracellular fluid. HHP eventually increases the number of certain food components generated by the interactions between substrates and enzymes [1,33]. These intercellular ruptures and turgor loss destruction accelerate intercellular diffusion and adjust the pH to the optimum condition for the specific enzymatic reaction.

Pressure and time combinations of 400 MPa and 30 min generated higher concentrations of pinitol, sucrose, stachyose, and raffinose (Figure 1). Although the metabolic pathways of these oligosaccharides and sugar alcohol are complicated, the enzymatic reactions are determined by reaction and diffusion rates. This unique non-linear concentration gradient could have resulted from the intercellular diffusion of precursor substrates to generate these oligosaccharides and sugar alcohol. Another possible mechanism was the denaturation of enzymes involved in the generation of these oligosaccharides and sugar alcohol. In our previous study, the cellular structures of soybeans were partly destroyed by HHP at 200–400 MPa, similar to freeze-thawed soybeans [13,34]. Maximal mass transfer occurred at 200–400 MPa due to the destroyed cellular structures allowing for increased substrate diffusion at higher pressure levels.

In contrast, enzymatic proteins were slightly denatured at specific pressure and time combinations, especially over 400 MPa for 30 min, and the apparent enzymatic reaction rate decreased, reducing the concentrations of certain compounds (Figure 1 and Figure 2). In our previous study, we investigated the enzymatic activity of extracted glutamate decarboxylase (GAD)—catalyzes the reaction from glutamic acid to γ-aminobutyric acid (GABA)—which was maintained even after applying HHP of 200 MPa [11]. Additionally, GAD activity decreased at higher pressure [11]. Hulle reported that the enzymatic activity of pectin methylesterase (PME) in aloe vera juice was a function of the pressure, pressuring time, and pH [35]. Furthermore, some physicochemical properties in aloe vera juice showed non-linear behavior over the selected range of process conditions. Because non-covalent bonds can be disrupted and reformed based on external factors such as ionic strength, pH, temperature, and pressure, HHP has been utilized to modify various protein properties, including enzymatic reactions [36]. Some researchers have investigated pressure–time relationships of enzymatic reactions and found that proteins can dissolve or precipitate upon application of high pressure during the denaturation process [37,38]. These changes are generally reversible in the pressure range of 100–300 MPa and irreversible at pressures higher than 300 MPa. Denaturation may be due to the destruction of hydrophobic and ion pair bonds and the unfolding of molecules. Although the effects of the combination of pressure and temperature on plant-based foods were not directly applicable to explain our observations, they may improve our understanding.

Overall, in the pressure range of 100–300 MPa, diffusion rate is the main rate-limiting step in the production of oligosaccharides, sugar alcohol, and soyasapogenol A, while at higher pressures or specific combinations of lower pressures and extended pressuring times resulting in protein denaturation, enzymatic inactivation is the overall rate-limiting step (Figure 1 and Figure 2). A deeper understanding of the metabolomics of pressurized soybeans will provide important insights into the complex enzymatic reaction system. Furthermore, soybean protease functions to increase the average hydrophobic value of free amino acids, and the hydrophobic amino acid residues with high bitterness values are removed to reduce bitterness [9]. As a result, HHP expedited some enzymatic reactions and chemical alternations, enriching pinitol and free amino acids and reducing soyasapogenol A, isoflavones, and oligosaccharides in pressurized soybeans compared to thermal processing [39]. Yang reported that ultrasound treatment markedly decreased isoflavone, daidzin, and genistin contents, whereas daidzein and genistein contents increased compared with the untreated sample [40]. Trypsin inhibitors in soybeans are widely known as anti-nutritious ingredients. HHP affects the activities of trypsin inhibitors by destroying noncovalent and disulfide bonds in proteins [41]. Trypsin inhibitor content was reduced to 22.3% and 10.76% by germination and applying HHP at 450 MPa for 5 min [42]. Applying HHP to soybeans leads to the inactivation of trypsin inhibitors at higher pressures; therefore, certain proteases react with the substrate, reducing anti-nutritious ingredients [43]. Since soybean protease has been shown to reduce bitterness and improve oligopeptides’ functional properties, additional functional property tests of pressurized soybeans would be beneficial. 

Legumes generally require thermal processing to promote digestibility and inactivate undesirable components, resulting in lower nutrition and functionality. Laguna reported that HHP improved the gastric digestibility of pea protein [44]. In contrast, high-pressure isolation of red kidney bean proteins resulted in significantly lower in vitro digestibility by trypsin [45]. Soluble compounds, such as pinitol, drastically decreased after boiling; therefore, non-thermal-pressurized soybeans enriched in low molecular weight components such as pinitol and free amino acids, and with decreased Group A soyasapogenol and isoflavones have a high potential of being a value-added food source with better taste. After applying HHP, soybeans were partly destroyed, leading to the softening of their texture [15,46]. Therefore, soybeans treated under HHP at higher pressures of 500–600 MPa with no bitter taste were soft enough to eat, and it was not necessary to subject them to further thermal cooking after applying HHP. In the future, investigations of functional properties and inactivation behavior of microbes from the viewpoint of food safety [47] and the health benefits of HHP-treated soybeans will be essential for its further use in food industries.

## 5. Conclusions

The distributions of oligosaccharides, pinitol, soyasapogenol A, and fatty acids in pressurized soybeans (cv. Yukihomare) were investigated. At higher pressures, HHP decreased the amount of soyasapogenol A detected in this study and that of isoflavones detected in our previous work, thus reducing the bitter–astringent taste. Although other processing methods can also modify the physicochemical properties of soybeans, HHP enables the enrichment of specific nutrients and inactivates microbes simultaneously. Nutrient enrichment via HHP is a more rapid and cost-effective alternative to breeding plants. If the added value to consumers justifies the cost, HHP-treated food production will be significantly promoted.

Through oxidation and enzymatic reactions, HHP influences taste-related compounds in soybeans, resulting in nutritious and functional value-added products.

## Figures and Tables

**Figure 1 foods-13-02214-f001:**
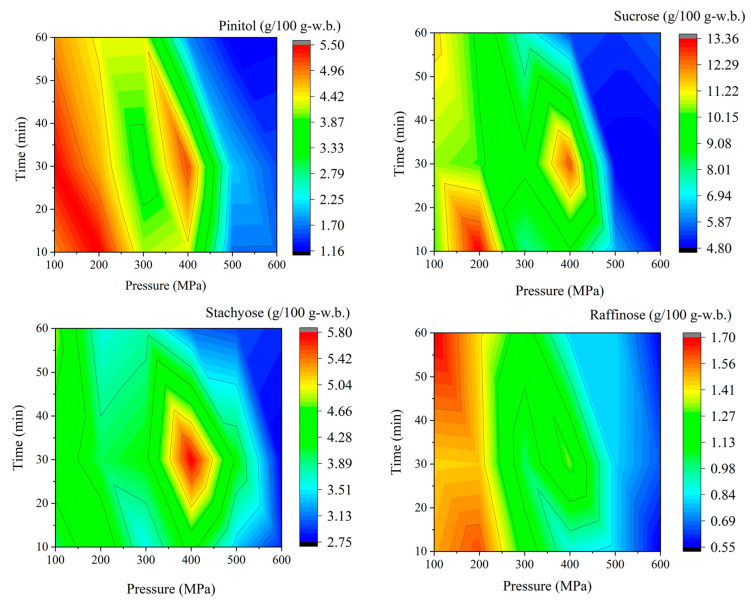
Contour diagram of pinitol, sucrose, stachyose, and raffinose concentrations in pressurized soybeans (cv. Yukihomare) (*n* = 3). The color grade indicates the concentrations (g per 100 g-w.b.) of soybean samples.

**Figure 2 foods-13-02214-f002:**
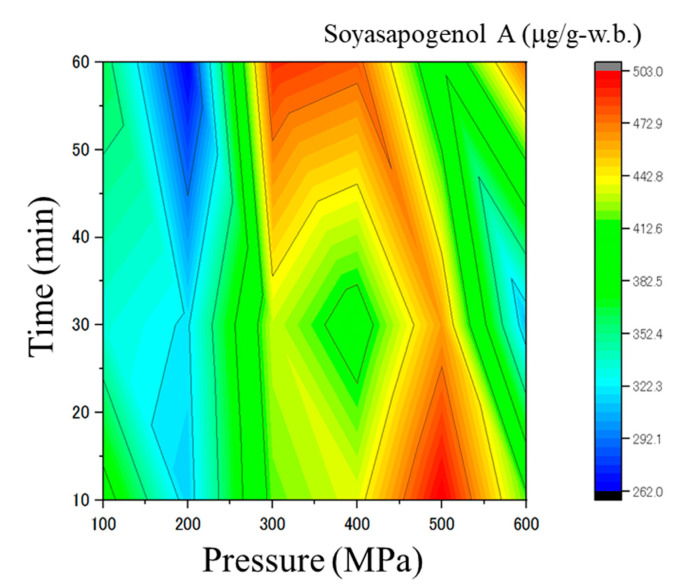
Contour diagram of soyasapogenol A concentrations in pressurized soybeans (cv. Yukihomare) (*n* = 3). The color grade indicates the concentrations (μg per one g-w.b.) in soybean samples.

**Figure 3 foods-13-02214-f003:**
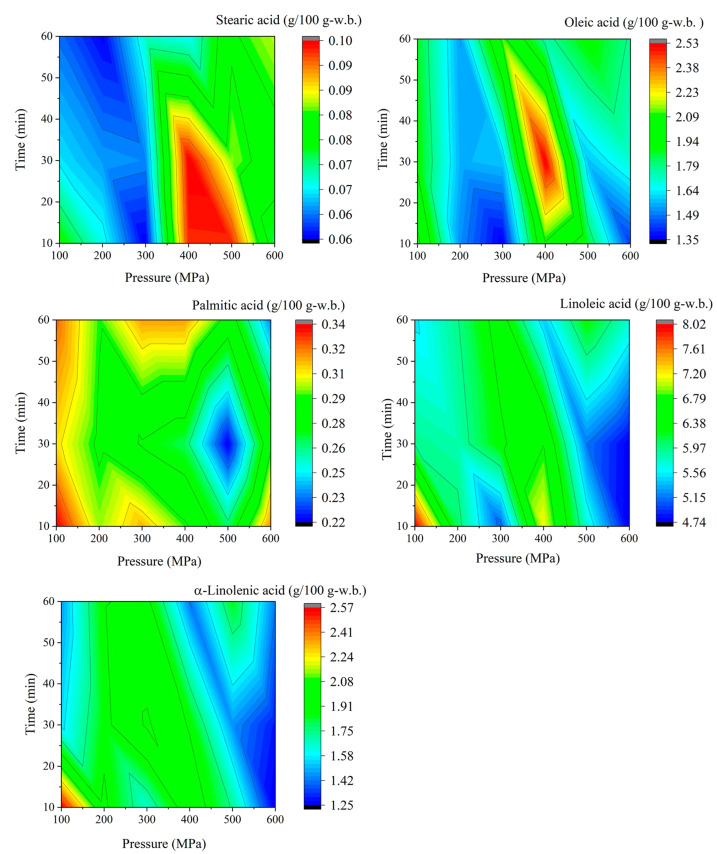
Contour diagrams of α-linolenic acid, linoleic acid, oleic acid, palmitic acid, and stearic acid in pressurized soybeans (*n* = 3). The color grade indicates the concentrations (g per 100 g-w.b.) of soybean samples (cv. Yukihomare).

**Table 1 foods-13-02214-t001:** Distributions of pinitol, sucrose, stachyose, raffinose, and soyasapogenol A in pressurized soybeans (cv. Yukihomare).

Pressure (MPa)	Time (min)	Pinitol (g/100 g)	Sucrose (g/100 g)	Stachyose (g/100 g)	Raffinose (g/100 g)	Soyasapogenol A (µg/g)
Untreated	Untreated	2.52 ± 0.53	14.69 ± 2.43	5.88 ± 1.27	1.82 ± 0.49	429.53 ± 13.07
100	10	4.86 ± 0.12	10.24 ± 1.01	3.99 ± 0.12	1.53 ± 0.32	394.48 ± 3.51
100	30	5.46 ± 0.37	10.61 ± 0.47	4.52 ± 0.19	1.45 ± 0.33	333.63 ± 0.57
100	60	4.84 ± 0.07	11.40 ± 0.26	4.79 ± 0.08	1.70 ± 0.02	362.55 ± 4.34
200	10	5.48 ± 0.07	13.35 ± 0.74	4.61 ± 0.69	1.60 ± 0.15	313.70 ± 8.51
200	30	4.71 ± 0.14	10.28 ± 0.60	4.02 ± 0.31	1.46 ± 0.05	320.32 ± 1.72
200	60	4.34 ± 0.08	9.97 ± 0.22	3.64 ± 0.18	1.43 ± 0.05	262.73 ± 9.88
300	10	4.13 ± 0.05	7.98 ± 0.10	3.58 ± 0.09	1.22 ± 0.17	419.61 ± 5.85
300	30	3.65 ± 0.42	9.29 ± 0.24	4.22 ± 0.21	1.02 ± 0.23	431.98 ± 3.09
300	60	4.33 ± 0.34	7.39 ± 0.20	3.75 ± 0.25	1.20 ± 0.05	490.37 ± 6.29
400	10	4.31 ± 0.71	9.0 ± 1.99	4.40 ± 0.78	0.88 ± 0.12	436.66 ± 4.02
400	30	5.15 ± 0.31	12.57 ± 0.04	5.78 ± 0.56	1.31 ± 0.10	400.52 ± 9.94
400	60	1.79 ± 0.13	5.54 ± 0.20	3.07 ± 0.25	0.8 ± 0.10	479.21 ± 5.21
500	10	1.68 ± 0.09	6.38 ± 0.16	3.37 ± 0.09	0.79 ± 0.02	502.89 ± 7.69
500	30	2.04 ± 0.18	4.81 ± 0.10	4.22 ± 0.21	0.81 ± 0.12	463.63 ± 3.13
500	60	1.16 ± 0.1	5.27 ± 0.00	2.97 ± 0.16	0.79 ± 0.02	386.49 ± 33.58
600	10	1.67 ± 0.30	4.85 ± 0.34	2.89 ± 0.33	0.55 ± 0.05	417.60 ± 51.59
600	30	1.52 ± 0.05	4.83 ± 0.17	2.75 ± 0.13	0.66 ± 0.04	308.32 ± 42.37
600	60	1.21 ± 0.02	5.72 ± 0.07	2.90 ± 0.05	0.56 ± 0.02	475.39 ± 19.47

The results are shown as mean ± SD (*n* = 3).

**Table 2 foods-13-02214-t002:** Distributions of pinitol, sucrose, stachyose, raffinose, and soyasapogenol A in boiled soybeans (cv. Yukihomare).

Boiling Time (min)	Pinitol (g/100 g)	Sucrose (g/100 g)	Stachyose (g/100 g)	Raffinose (g/100 g)	Soyasapogenol A (µg/g)
0	2.52 ± 0.53 a	14.69 ± 2.43 a	5.88 ± 1.27 a	1.82 ± 0.49 a	429.53 ± 13.07 a
10	3.43 ± 0.27 a	11.83 ± 0.25 a	4.62 ± 0.23 a	1.47 ± 0.08 a	363.17 ± 1.09 a
20	3.28 ± 0.31 a	12.03 ± 0.60 a	4.77 ± 0.12 a	1.36 ± 0.04 a	290.23 ± 0.94 a
30	3.47 ± 0.21 a	11.49 ± 0.53 a	4.24 ± 0.10 a	1.48 ± 0.18 a	313.19 ± 0.46 ac
40	4.24 ± 0.44 b	13.07 ± 0.41 a	5.18 ± 0.26 a	1.74 ± 0.09 a	321.26 ± 1.84 a
50	3.91 ± 0.21 b	13.52 ± 0.28 a	5.76 ± 0.45 a	1.72 ± 0.10 a	290.07 ± 3.09 b
60	3.82 ± 0.20 b	13.67 ± 0.59 a	5.01 ± 0.32 a	1.76 ± 0.05 a	279.15 ± 1.17 a

The results are shown as mean ± SD (*n* = 3). The different letters in the same compounds indicate significant differences (*p* < 0.05).

**Table 3 foods-13-02214-t003:** Distributions of α-linolenic acid, linoleic acid, oleic acid, palmitic acid, and stearic acid in pressurized soybeans (cv. Yukihomare). The results are shown as mean ± SD (*n* = 3).

Pressure (MPa)	Time (min)	a-Linolenic Acid (g/100 g-w.b.)	Linoleic Acid (g/100 g-w.b.)	Oleic Acid (g/100 g-w.b.)	Palmitic Acid (g/100 g-w.b.)	Stearic Acid (g/100 g-w.b.)
Untreated	Untreated	1.73 ± 0.35	5.62 ± 0.43	1.44 ± 0.15	0.31 ± 0.03	0.06 ± 0.01
100	10	2.57 ± 1.11	8.02 ± 2.08	2.12 ± 0.58	0.34 ± 0.04	0.08 ± 0.01
100	30	1.55 ± 0.25	5.95 ± 0.89	1.91 ± 0.11	0.31 ± 0.03	0.07 ± 0.01
100	60	1.47 ± 0.27	5.52 ± 0.82	1.87 ± 0.26	0.32 ± 0.01	0.06 ± 0.01
200	10	1.94 ± 0.87	6.07 ± 2.06	1.52 ± 0.11	0.30 ± 0.02	0.07 ± 0.00
200	30	1.87 ± 0.78	5.85 ± 1.59	1.56 ± 0.26	0.29 ± 0.05	0.07 ± 0.01
200	60	1.90 ± 0.60	5.96 ± 1.19	1.54 ± 0.08	0.29 ± 0.02	0.06 ± 0.01
300	10	1.65 ± 0.59	5.06 ± 1.20	1.35 ± 0.10	0.31 ± 0.03	0.06 ± 0.01
300	30	2.10 ± 0.88	6.31 ± 1.96	1.58 ± 0.19	0.28 ± 0.05	0.07 ± 0.01
300	60	1.95 ± 0.56	6.54 ± 1.20	2.12 ± 0.57	0.32 ± 0.05	0.07 ± 0.01
400	10	2.02 ± 0.89	7.17 ± 2.39	2.07 ± 0.17	0.29 ± 0.02	0.10 ± 0.04
400	30	1.89 ± 0.72	6.78 ± 2.18	2.53 ± 0.84	0.27 ± 0.01	0.10 ± 0.06
400	60	1.41 ± 0. 37	5.43 ± 1.18	1.76 ± 0.05	0.32 ± 0.02	0.07 ± 0.03
500	10	1.73 ± 0.92	5.73 ± 2.37	1.86 ± 0.45	0.26 ± 0.04	0.10 ± 0.05
500	30	1.45 ± 0.99	5.15 ± 2.83	1.58 ± 0.34	0.22 ± 0.03	0.09 ± 0.04
500	60	1.85 ± 0.51	6.29 ± 1.31	1.92 ± 0.31	0.29 ± 0.02	0.08 ± 0.03
600	10	1.25 ± 0.46	4.75 ± 1.07	1.42 ± 0.20	0.32 ± 0.04	0.08 ± 0.01
600	30	1.29 ± 0.33	4.76 ± 0.74	1.70 ± 0.42	0.29 ± 0.03	0.08 ± 0.05
600	60	1.42 ± 0.34	5.74 ± 1.28	1.78 ± 0.36	0.23 ± 0.01	0.09 ± 0.04

## Data Availability

The original contributions presented in the study are included in the article, further inquiries can be directed to the corresponding author.

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
