# Peer review of "Effects of High Hydrostatic Pressure on the Distribution of Oligosaccharides, Pinitol, Soysapapogenol A, and Fatty Acids in Soybean"

_foods, 2024, doi:10.3390/foods13142214_

Round 1

Reviewer 1 Report

Comments and Suggestions for Authors

The work represents a significant contribution to understanding the changes that occur in soybeans when exposed to High Hydrostatic Pressure (HHP). However, there are several aspects that need clarification and/or improvement.

Firstly, the specific soybean cultivar used must be indicated, as there is considerable variability among different varieties in the initial content of compounds of interest (oligosaccharides, lipids, proteins, etc.). This information should be provided in the Materials and Methods section and also mentioned in the conclusions, as the values obtained are specific to a particular cultivar.

Additionally, the moisture content of the soybeans should be specified, as the effect of HHP can vary depending on the water content of the seeds. Even if the seeds were purchased from a supermarket, it is known that post-harvest storage conditions and handling before processing can affect their moisture content.

The conclusions should consider that the improvements observed in soybean quality due to HHP could also be achieved through other means. For example, cultivars with a higher initial pinitol content could, after boiling, have values similar to those obtained with other cultivars and HHP. Alternatively, cultivars exposed to controlled pre-harvest stressing conditions (draught, pests...) might have higher pinitol values.

It should also be noted that using HHP entails an additional cost, which should be justified by the improvements in nutritional quality. A paragraph related to that must be added.

Finally, it should be mentioned that while HHP-treated soybeans show improvements compared to water-boiled soybeans, the latter can still be directly consumed. It is not specified whether HHP-treated soybeans would require further treatment to achieve an organoleptic quality suitable for consumption. Did the HHP affect the soybean texture? If additional treatment (e.g., thermal) is necessary after HHP, the benefits gained could be diminished in this new context.

 Other minor details:

Line 37: delete the “t”

In the contour diagrams, please increase the font size of the colour scale. The current font size is very small and almost impossible to read.

Comments on the Quality of English Language

The English language is good and appropriated for a scientific publication.

Author Response

Dear Reviewer 1

We truly appreciate the constructive comments given by the reviewer. After
careful consideration, we revised the manuscript according to the reviewer's
comments and hope it may be more generally favorable for possible publication in
Foods. Our responses are listed point-by-point as follows (changes shown in revised
manuscript in yellow):

>Firstly, the specific soybean cultivar used must be indicated, 
>as there is considerable variability among different varieties 
>in the initial content of compounds of interest (oligosaccharides,
>lipids, proteins, etc.). This information should be provided in 
>the Materials and Methods section and also mentioned in the 
>conclusions, as the values obtained are specific to a 
>particular cultivar.

Thank you for your comment. I add the information about cultivar of
the soybean as follows from the introduction, the materials and 
methods, results, figure legends, table legends, discussions, and 
conclusions.

L15
A as taste ingredients in soybean (Glycine max (L.) Merr.)(cv. Yukihomare) were evaluated.

Lines 87 to 89.
Soybeans (cv. Yukihomare) harvested in Hokkaido, Japan, in November 2021 and October 2022 (Asahishokuhin Corp., Kobe, Japan) were purchased from a local super-market. 

L101
Approximately 20 g of dried soybeans (cv. Yukihomare) were soaked in 60 g of ul-trapure water for 15 hours

L127
One gram of frozen soybeans (cv. Yukihomare) was cut into small particles with a knife, and then soybean particles with 5 g of ultrapure water were homogenized with Polytron MR2100 (Kinematica, Malters, Switzerland) at 12000 rpm for 5 min at 4°C.

L149
Soybean samples (cv. Yukihomare)were frozen at -20°C for 24 h and were continuous-ly vacuumed for 24 h.

L183
Ten g of untreated and pressurized soybeans (cv. Yukihomare) were cut into small particles with a knife and were mixed with 60 ml of chloroform-methanol (2:1, v/v). 

L221
The distributions of pinitol and three oligosaccharides, sucrose, stachyose, and raffinose were measured in untreated and pressurized soybeans(cv. Yukihomare) (Table 1). 

L241
Figure 1. Contour diagram of pinitol, sucrose, stachyose, and raffinose concentrations in pressurized soybeans(cv. Yukihomare) (N=3).

L251, Table 1 
Distributions of pinitol, sucrose, stachyose, raffinose, and soyasapogenol A in pressurized soybeans (cv. Yukihomare)

L259
The distributions of soyasapogenol A as an astringent/bitter taste component were measured in untreated and pressurized soybeans (cv. Yukihomare) (Table 1). 

L274
The distributions of pinitol, sucrose, stachyose, raffinose, and soyasapogenol A in boiled soybeans (cv. Yukihomare) were also measured (Table 2). 

Table 2 Distributions of pinitol, sucrose, stachyose, raffinose, and soyasapogenol A in boiled soybeans (cv. Yukihomare)

L324-325
Figure 2. Contour diagram of soyasapogenol A concentrations in pressurized soybeans (cv. Yukihomare) (N=3).

L330-31
The distributions of major fatty acids such asa-linolenic acid, linoleic acid, oleic acid, palmitic acid, and stearic acid, were measured in untreated and pressurized soy-beans (cv. Yukihomare) (Table 3).

L348
Table 3  Distributions of a-linolenic acid, linoleic acid, oleic acid, palmitic acid, and stearic acid in pressurized soybeans (cv. Yukihomare)

L351
Figure 3. Contour diagram of a-linolenic acid, linoleic acid, oleic acid, palmitic acid, and stearic acid in pressurized soybeans (N=3). Color grade indicates the concentration g per 100 g-w.b. of soybean samples (cv. Yukihomare)

L380
The application of HHP to soybeans(cv. Yukihomare) at 100 to 600 MPa for 10 to 60 min resulted in a higher accumulation of pinitol and a reduction of oligosaccharides at the specific pressure–time combinations. 

L554
Distributions of oligosaccharides, pinitol, soyasapogenol A, and fatty acids in pressur-ized soybeans (cv. Yukihomare) were notably investigated. 

>Additionally, the moisture content of the soybeans should be 
>specified, as the effect of HHP can vary depending on the water 
>content of the seeds. Even if the seeds were purchased from a
>supermarket, it is known that post-harvest storage conditions 
>and handling before processing can affect their moisture content.

Authors add the information about the moisture content of dried soybean
and water-soaked soybeans. In this present study, prior to HHP,
soybean samples fully uptake a soaking water and  
soybeans with saturated moisture content are used in high-pressure experiments.

L102-104
The moisture content of dried soybean before water-soaking was approximately 0.22 g-water/g-d.s., and that of water-soaked soybean was approximately 1.90 g-water/g-d.s. In this present study, all analyses were carried out with the water-soaked soybeans. 

>The conclusions should consider that the improvements observed in 
>soybean quality due to HHP could also be achieved through other means. 
>For example, cultivars with a higher initial pinitol content could, 
>after boiling, have values similar to those obtained with other
> cultivars and HHP. Alternatively, cultivars exposed to controlled 
>pre-harvest stressing conditions (draught, pests...) might have 
>higher pinitol values.
>It should also be noted that using HHP entails an additional cost, 
>which should be justified by the improvements in nutritional quality. 
>A paragraph related to that must be added.

Thank you for the good comments. HHP is gradually known for the 
better nonthermal processing for food manufacturing these days.
In Japan, China, EU countries and the US, the HHP products are widely
sold especially in the field of rice products, juice, and cooked foods.
However, the cost of these HHP-products are a little bit expensive
than the traditional processed products.
I add the explanation of merits and demerits of HHP including the cost and comparison with other processing in Conculsions as follows: 

L556-562
Although other processing could modify the physico-chemical properties of soybeans, HHP enables to enrichment of the specific nutrition and inactivates microbes, simultaneously. 
While breeding the plants takes a long time, nutrient enrichment by HHP takes less time and is more cost-effective. On the other hand, the price per product may increase.If it becomes possible to provide more value to consumers than the cost, it can be expected to contribute greatly to the food supply.

>Finally, it should be mentioned that while HHP-treated soybeans 
>show improvements compared to water-boiled soybeans, the latter can 
>still be directly consumed. It is not specified whether HHP-treated 
>soybeans would require further treatment to achieve an organoleptic
> quality suitable for consumption. Did the HHP affect the soybean
> texture? If additional treatment (e.g., thermal) is necessary 
>after HHP, the benefits gained could be diminished in this new 
>context.
After HHP at higher pressure levels, the texture of soybean samples
were similar to those after boiling.
In introduction, I add the text about the HHP-assited softning the 
texture as shown below.

L51-57
Dried soybean is generally edible after water-soaking and boiling, which leads to softening a texture and a reduction of taste-related compounds such as lipoxygenase. On the other hand, certain enzymatic reactions have occurred, which resulted in changes in ingredients such as proteins, polysaccharides, and lipids during the processing and storage of soybeans. HHP also made it possible to soften a texture and enrich free amino acids via a quick enzymatic reaction at the optimum pressure-time combinations, which was especially beneficial for the nutritional components of soy-beans [11-13].

L545-550
 After HHP, the soybean was partly destroyed, which led to the softening of the tex-ture. Therefore, HHP-treated soybeans at higher pressure levels of 500-600 MPa are edible, and it was not necessary to further thermal cooking. In the future, the investi-gations of functional properties and health benefits of HHP-treated soybeans will be essential.

 Other minor details:

Line 37: delete the “t”
>> I delete the "t" in line 37.

>In the contour diagrams, please increase the font size of the colour scale. The current font size is very small and almost impossible to read.
Authors improve the contour diagram in Figure 1 and Figure 3 .
Font sizes of these colour scales were expanded from 14 pt to 28 pt.

Reviewer 2 Report

Comments and Suggestions for Authors

The manuscript provides interesting information about the impact of HHP on soybean. Please see my detailed comments below.

- I believe it's worth mentioning the effect of HHP on enzyme inhibition in the Introduction. 

- Please describe in detail the HHP and the thermal treatment; what was the weight of the samples, what was the max temperature reached during the treatments, how did you monitor it?

- How many biological and how many technical replicates did you perform?

- Some of the cited literature -especially in the Discussion - is quite outdated. I'd advise the Authors to enrich the Discussion with more recent literature.

- The Conclusion section should focus on the findings of the current study. Please rephrase.

Comments on the Quality of English Language

Although the quality of the language is good there are some parts with phrasing issues and a few typos. For instance, lines 37-39, 43-44, 366, 401-402, 444, 450. I'd advice the Authors to proofread the manuscript again.

Author Response

Dear Reviewer 2

We truly appreciate the constructive comments given by the reviewer. After careful consideration, we revised the manuscript according to the reviewer's comments and hope it may be more generally favorable for possible publication in Foods. Our responses are listed point-by-point as follows (changes shown in revised manuscript in yellow):

>I believe it's worth mentioning the effect of HHP on enzyme inhibition 
>in the Introduction. 

Thank you for good suggestion. I add the explanation of the inhibition the enzymatic oxidation such as polyphenol oxidase from green tea with the references.

L59-61,
 HHP is also regarded as a promising technology with minimal impact on food products’sensory attributes, promoting inhibition of the enzymatic oxidation catalyzed by specific enzymes such as polyphenol oxidase [5, 14-16]

- Please describe in detail the HHP and the thermal treatment; what was 
>the weight of the samples, what was the max temperature reached 
>during the treatments, how did you monitor it?
Fifty grams of soybeans were applied to each experiment such as HHP, thermal treatment, and untreated. Temperature in the HHP-vessel was maintained by the circulating water in the outer shell of the vessel. The temperature of the vessel was monitored by thermocouple. Authors add the explanation as follows:

L105-109
After water soaking, approximately 50 g of soybeans were wiped with a paper towel and vacuum-packed in polyethylene pouches (Eiken Kizai, Tokyo, Japan) using a vacuum sealer (Furukawa Seisakujo, Tokyo, Japan). Soybean samples in pouches without water were placed in a stainless-steel vessel of an HHP device (MFP7000; Mitsubishi Heavy Industry, Tokyo, Japan).

- How many biological and how many technical replicates did you perform?

Authors add the explanation in lines 120 to 121 as follows: 
At least 15 biologically different soybean grains were used for the single analysis, and then all quantification analyses were carried out in triplicate. 

>- Some of the cited literature -especially in the Discussion - is quite outdated. I'd advise 
>the Authors to enrich the Discussion with more recent literature.
According to reviwer's comment, authors modify the more recent literatures and improve the texts as follows. Several references were still remained with it's importance to discuss

L405-407
 Because D-pinitol has a role in drought and high salinity stress, as well as heat-induced water deficits in legumes, pinitol concentrations were raised under stress conditions [25]. The functions of D-pinitol are antioxidant, anti-inflammatory, and an-ti-carcinogenic properties, therefore, high accumulation of D-pinitol leads to high val-ue-added soybean [26]. 

L412-416
Furthermore, the functions of D-pinitol are antioxidant, anti-inflammatory, and an-ti-carcinogenic properties [26], and D-pinitol improves diabetic sarcopenia [27]. D-pinitol also increased the expression of the specific protein that has functions of an-ti-inflammation and antiapoptosis [27]. 

L420-421
To increase the retention and utilization of soybean components, antinutritional fac-tors in soybean meal must be inactivated [28].

L504-509
 Because non-covalent bonds can be disrupted and reformed based on external factors such as ionic strength, pH, temperature, and pressure, HHP has been utilized to modi-fy various protein properties including enzymatic reactions [36]. Some researchers have investigated the pressure-time relationships for enzymatic reactions that the proteins may dissolve or precipitate on applying high pressure during the denatura-tion process [37, 38].

25.    Phang H, Shao G, Lam M. Salt Tolerance in Soybean. Journal of Integrative Plant Biology, 2008, 50(10), 1196-1212. https://doi.org/10.1111/j.1744-7909.2008.00760.x

26.    KuÅŸ, N. Åž. Biological Properties of Cyclitols and Their Derivatives. Chemistry & Biodiversity, 2023. 21(1), e202301064. https://doi.org/10.1002/cbdv.202301064 

27.    Xin Yu, Pei Li, Baoying Li, Fei Yu, Wenqian Zhao, Xue Wang, Yajuan Wang, Haiqing Gao, Mei Cheng, and Xiaoli L, D-Pinitol Improves Diabetic Sarcopenia by Regulation of the Gut Microbiome, Metabolome, and Proteome in STZ-Induced SAMP8 Micei. Journal of Agricultural and Food Chemistry. 2024, 72 (25), 14466-14478. doi: 10.1021/acs.jafc.4c03929

28.    Huang L, Xu Y Effective reduction of antinutritional factors in soybean meal by acetic acid-catalyzed processing. Journal of Food Processing and Preservation, 2018, 42(11), e13775. https://doi.org/10.1111/jfpp.13775 

36.    Braspaiboon, S.; Laokuldilok, T. High Hydrostatic Pressure: Influences on Allergenicity, Bioactivities, and Structural and Functional Properties of Proteins from Diverse Food Sources. Foods 2024, 13, 922. https://doi.org/10.3390/foods13060922 

37.Zhang, W., Li, X., Wang, X., Li, H., Liao, X., Lao, F., Wu, J., & Li, J. Decoding the Effects of High Hydrostatic Pressure and High-Temperature Short-Time Sterilization on the Volatile Aroma Profile of Red Raspberry Juice. Foods, 2023, 13(10), 1574. https://doi.org/10.3390/foods13101574

- The Conclusion section should focus on the findings of the current study. Please rephrase.
As reviwer pointed out, I delete the first two lines from the conclusion.

>Comments on the Quality of English Language
>Although the quality of the language is good there are some parts with phrasing issues and a few typos. For instance, lines 37-39, 43-44, 366, 401-402, 
>444, 450. I'd advice the Authors to proofread the manuscript again.

Thank you for your comments. Authors correct the phrasing and a few typos.
L37-39, Authors delete "t".
L43-44
Recently in Japan, soymilk cream has been widely used as an alternative to cow milk. On the other hand, in Western countries, soybeans are mainly processed into soybean meal and seed oil [10].

L383-385(originalL366) 
 Among sugars in soybean, sucrose was the biggest component (48.90%, 50.65 mg/g), followed by stachyose (39.26%, 40.68 mg/g) and raffinose (9.38%, 9.72 mg/g), with glucose and fructose contributing a very little part (2.46%) [22]. 

L420-421 (original L401-402)
To increase the retention and utilization of soybean components, antinutritional factors must be inactivated [28].

L461-463 (original444)
 The synthesis of these major fatty acids in soybeans has been reported to be catalyzed by ω-6 fatty acid unsaturation enzyme (FAD2) and ω-3 fatty acid unsaturation enzyme (FAD3) [10]. (insert -)

L469-471 (original L450)
The mechanisms of HHP-induced nutritional changes were firstly associated with oxidation in the cellular biomaterials, resulting in secondary metabolites at lower pressure levels [12]. 

Round 2

Reviewer 2 Report

Comments and Suggestions for Authors

No further comments. The manuscript is now ready for publication.

Comments on the Quality of English Language

Even though the quality of the language is good some minor editing is still needed prior to publication.

Author Response

Dear Reviewer 2

We appreciate your comments.  Aftercareful consideration, we revised the manuscript according to the reviewer's comments again, and hope it may be more favorable for possible publication in Foods.

I modify the  texts more clearly for better understanding.  The modified texts were colored with blue  for the second revision, and the first revisions with yellow color were also maintained.
From the editor's comments, I newly add references, and some self citations and  references from Foods were deleted and replaced to the new references (refs, 2,5,12, 18, 38, 44, 46,47).

In Table2, result about stacyose for boiling time of 0 min in the 1st revised version were not correctly located, then I correct the position (5.88+1.27a).